# Learning Velocity Prior-Guided Hamiltonian-Jacobi Flows with Unbalanced Optimal Transport

## Abstract

The connection between optimal transport (OT) and control theory is well established, most prominently in the Benamou–Brenier dynamic formulation. With quadratic cost, the OT problem can be reframed as a stochastic control problem in which a density $\rho_t$ evolves under a controlled velocity field $v_t$ subject to the continuity equation $\partial_t \rho_t + \nabla \cdot (\rho_t v_t) = 0$. In this work, we introduce a velocity prior into the continuity equation and derive a new Hamilton–Jacobi–Bellman (HJB) formulation to learn dynamical probability flows. We further extend the approach to the unbalanced setting by adding a growth term, capturing mass variation processes common in scientific domains such as cell proliferation and differentiation. Importantly, our method requires training only a single neural network to model $v_t$, without the need for a separate model for the growth term $g_t$. Finally, by decomposing the velocity field as $v_{\text{total}} = v_{\text{prior}} + v_{\text{corr}}$, our approach is able to capture complex transport patterns that prior methods struggle to learn due to the curl-free limitation.

## 1 Introduction

From flow matching (FM) to action matching (AM), learning transport maps between distributions has been widely explored in recent years (Lipman et al., 2022; Albergo and Vanden-Eijnden, 2022; Liu et al., 2022; Neklyudov et al., 2023a). *Flow Matching (FM)* (Lipman et al., 2022) learns a time–dependent velocity field $u_t$ that pushes $\rho_0$ to $\rho_1$ and can realize highly expressive transport paths; however, the original FM with independent coupling between source and target does not guarantee *least action* by minimizing the kinetic energy in the Benamou–Brenier sense. Instead, it trains $u_t$ to match conditional expectations of displacement vectors under a chosen interpolation scheme, which may yield non-optimal flows.

*Action Matching (AM)* (Neklyudov et al., 2023a) addresses this by parameterizing a scalar potential $s_t$ whose gradient $\nabla s_t$ induces the transport, aligning with the optimality conditions of OT and yielding lower kinetic energy than unconstrained FM. The price is reduced expressiveness: $\nabla s_t$ is *curl-free*, so AM cannot directly represent rotational or cyclic dynamics that are common in scientific domains. From the Helmholtz decomposition perspective (Neklyudov et al., 2023a), any vector field $u_t^*$ can be written as $u_t^* = \nabla s_t^* + w_t$ with $w_t$ divergence-free (Ambrosio et al., 2005, §8.4.2). Under this lens, AM retains only the gradient component and discards discards $w_t$, explaining both its energy efficiency and its inability to encode rotations and cycles.

In this paper, we seek a middle ground – expressive like FM, energy-aware like AM – by introducing a velocity prior $v_{\text{prior}}$ and learning only the residual potential. We note that even compared with energy-aware FM variants such as OT–CFM(Pooladian et al., 2023; Tong et al., 2023a), our approach achieves better energy efficiency, as demonstrated in upper Table 1. Specifically, we decompose the velocity field as $v_{\text{total}}(t, x) = v_{\text{prior}}(t, x) + \nabla s_t(x)$. Here $v_{\text{prior}}$ captures known rotational dynamics or domain-specific effects such as RNA velocity in single-cell biology, while $\nabla s_t$ accounts for the OT-consistent gradient component. We train $s_t$ by minimizing a modified Hamilton–Jacobi residual that incorporates the prior, together with boundary terms that ensure $\rho_0 \to \rho_1$. This *residualized* design preserves OT optimality conditions for the learned component, improves interpretability,

Table 1: Least Action Comparison for Balanced Gaussian Translation

| Method | Mean error | Cov. error | $\mathbf{W_2}$ | Control action $(\nabla s)$ | Total kinetic $(v_{\text{total}})$ |
|---|---|---|---|---|---|
| Flow Matching | 0.204 | 0.804 | 0.582 | 18.369 | 18.369 |
| OT-FM | 0.149 | **0.659** | **0.402** | 18.707 | 18.707 |
| VP-HJF (Ours) | 0.102 | 0.791 | 0.577 | **0.624** | **17.955** |
| Prior-only ($\alpha{=}1$) | **0.008** | 0.171 | 1.351 | 0 | 36.250 |

and injects inductive bias without paying the kinetic-energy cost of unconstrained original FM. We name our approach the Velocity Prior Hamiltonian-Jacobi Flow (VP-HJF).

**Motivation in practice**    In domains like single–cell biology and physical systems with known drifts, accurate priors are available yet incomplete and and mass change such as cell proliferation and decay is ubiquitous. VP–HJF exploits these priors to encode hard–to–learn structure. The residual potential learns both the correction that the prior can not explain and the mass changes through the growth term. This yields a compact and interpretable alternative to fully free vector–field models, particularly effective when local supervision is noisy but prior knowledge is rich.

## 2  BACKGROUND

**Dynamical Optimal Transport**    Beyond the classic static Monge–Kantorovich formulation in OT (Ambrosio et al., 2005; Villani et al., 2008), there exists a dynamical formulation known as the Benamou–Brenier problem which links OT with PDEs by representing the $W_2$ distance as the minimum kinetic energy where $\rho_t$ is density and $v_t$ is a velocity field with boundary conditions: $\rho_{|t=0} = \rho_0, \; \rho_{|t=1} = \rho_1$, (Benamou and Brenier, 2000):

$$W_2^2(\rho_0, \rho_1) = \inf_{\rho_t, v_t} \int_0^1 \int \tfrac{1}{2}\|v_t(x)\|^2 \rho_t(x)\, dx\, dt, \quad \partial_t \rho_t + \nabla \cdot (\rho_t v_t) = 0. \tag{1}$$

**Unbalanced Optimal Transport**    When total mass change over time such as following a growth-decay process in biology, we add a growth rate $g_t(x)$ term to the continuity equation to incorporate the weight changes (Chizat et al., 2018):

$$\partial_t \rho_t(x) + \nabla \cdot \big(\rho_t(x)\, v_t(x)\big) = g_t(x)\, \rho_t(x), \quad \rho_{|t=0} = \rho_0, \; \rho_{|t=1} = \rho_1, \tag{2}$$

The Wasserstein–Fisher–Rao distance with scale $\delta > 0$ is defined as the minimal action balancing transport cost and mass change:

$$\mathrm{WFR}_\delta^2(\rho_0, \rho_1) = \inf_{\rho, v, g} \int_0^1 \int \left( \tfrac{1}{2}\|v_t(x)\|^2 + \tfrac{\delta^2}{2} g_t(x)^2 \right) \rho_t(x)\, dx\, dt, \quad \text{s.t. } Eq.2. \tag{3}$$

**Hamilton–Jacobi–Bellman (HJB)**    We recall the classical connection between optimal control and Hamilton–Jacobi (HJ) theory. Consider a deterministic control system with state $x(t) \in \mathbb{R}^d$, control $u(t)$, dynamics $\dot{x} = f(x, u, t)$, running cost $L(x, u, t)$, and terminal cost $\psi(x)$. The *value function*

$$V(t, x) \;=\; \inf_{u(\cdot)} \left\{ \int_t^1 L\big(x(s), u(s), s\big)\, ds \;+\; \psi\big(x(1)\big) \right\}$$

gives the minimal cost-to-go from $(t, x)$ under admissible controls. It is well known that $V$ solves the Hamilton–Jacobi–Bellman (HJB) equation $\partial_t V(t, x) \;+\; H\big(x, \nabla V(t, x), t\big) \;= 0$, where $V(1, x) = \psi(x)$ and the Hamiltonian is $H(x, p, t) \;:=\; \inf_u \left\{ L(x, u, t) + p^\top f(x, u, t) \right\}$ with $p = \nabla V(t, x)$.

**Action Matching (AM)**    AM fits a scalar potential $s_\theta$ to learn a energy-minimizing flow between distributions by minimizing the (un)balanced HJB residuals.

$$\mathcal{L}_{\text{uAM}} = \int_0^1 \mathbb{E}_{x \sim \rho_t} \left[ \partial_t s_\theta(t, x) + \tfrac{1}{2}\|\nabla_x s_\theta(t, x)\|^2 + \tfrac{1}{2} s_\theta^2(t, x) \right] dt, \tag{4}$$

with boundary constraints as: $\mathbb{E}_{x \sim \rho_0}[s_0(x)] - \mathbb{E}_{x \sim \rho_1}[s_1(x)]$.

## 3 METHODOLOGY

We introduce a velocity-prior guided approach, the *Velocity Prior Hamiltonian–Jacobi Flow (VP-HJF)*, to solve the unbalanced optimal transport problem under the Wasserstein–Fisher–Rao (WFR) metric (Eq. 2, 3). In contrast to prior approaches that fit two separate networks—one for transport and one for growth (Zhang et al., 2024; Wang et al., 2025), our method trains a single neural network. Following (Neklyudov et al., 2023a), we can represent both the transport velocity field and the growth term through a single scalar potential.

**Proposition 3.1** (Neklyudov et al., 2023a, Prop. 3.3). *Suppose we have a continuous dynamic flow with density $\rho_t$. Under mild conditions, there exists a unique scalar potential function $\hat{s}_t(x)$ such that the unbalanced continuity equation (2) is satisfied, with the velocity field and growth function given by $v_t^*(x) = \nabla \hat{s}_t(x), g_t^*(x) = \hat{s}_t(x)$.*

Building on Proposition 3.1, we reduces the WFR problem to learning a single model and incorporate problem-specific dynamics through a simple velocity decomposition. Specifically, we decompose the velocity field into two parts: a known velocity prior and a learnable corrective velocity field component:

$$v_{\text{total}}(t, x) = v_{\text{prior}}(t, x) + v_{\text{corr}}(t, x), \tag{5}$$

where $v_{\text{prior}}$ encodes domain knowledge (e.g. translations, rotations, RNA velocity), and $v_{\text{corr}}$ is the data–driven corrective component. In this way, the prior captures coarse dynamics while the model focuses on refinements such as correcting the residual transport and learning mass imbalance that the prior cannot explain. In essence, our approach improves interpretability and reduces the learning complexity through adding prior knowledge of the velocity field $v_{\text{prior}}$ – leaving the learnable velocity field $v_{\text{corr}}$ simpler learning tasks compared with other generative modeling methods of learning the entire velocity field $v_{\text{total}}$. Intuitively, our approach pays kinetic cost only for the *correction* to the prior drift and for the mass growth-decay component, making learning more efficient.

We can now define our velocity-prior guided unbalanced OT problem under the least-action principle as:

**Definition 3.2.** Consider the following least-action objective with $\delta = 1$ and subject to the unbalanced *velocity-prior guided* continuity equation :

$$\mathcal{A}(\rho, v_{\text{corr}}, g) = \int_0^1 \int \left( \tfrac{1}{2} \|v(t, x)\|^2 + \tfrac{1}{2} g(t, x)^2 \right) \rho_t(x) \, dx \, dt, \tag{6}$$

$$\text{s.t.} \quad \partial_t \rho_t = -\nabla \cdot \left( \rho_t \left( v_{\text{prior}} + v_{\text{corr}} \right) \right) + g_t \, \rho_t, \quad \rho_{|t=0} = \rho_0, \ \ \rho_{|t=1} = \rho_1. \tag{7}$$

Note that in our method we do *not* optimize over $\rho$ directly. Instead, $\rho_t$ is *induced* by a parametric flow $\Phi_t^\theta$ via $\dot{x} = v_{\text{prior}}(t, x) + \nabla s_\theta(t, x)$ and defined as $\rho_t^\theta = (\Phi_t^\theta)_{\#} \rho_0$.

**Prior-guided HJB residual** Since solving for the minimum-action problem in primal form in Definition 3.2 is intractable, we turn to its dual formulation. The key derivation step is to introduce a scalar potential $s(t, x)$ as the Lagrange multiplier for the prior-guided continuity equation and applying the Fenchel–Young inequalities to the velocity field and growth term. We then obtain the following dual lower bound (see Appendix A for details):

$$\mathcal{A}(\rho, v_{\text{corr}}, g) \geq \mathbb{E}_{\rho_0(x)} [s_0(x)] - \mathbb{E}_{\rho_1(x)} [s_1(x)]$$
$$- \int_0^1 \int \rho_t(x) \left( \partial_t s + \tfrac{1}{2} \|\nabla s\|^2 + \nabla s \cdot v_{\text{prior}} + \tfrac{1}{2} s^2 \right) dx \, dt. \tag{8}$$

The bound is tight point-wise if and only if when we choose the primal variables as

$$v_{\text{corr}}(t, x) = \nabla_x s(t, x), \qquad g(t, x) = s(t, x),$$

which shows that $s$ can simultaneously control both the *corrective* transport $\nabla_x s$ and the local growth $s$. We can then plug these back to the continuity equation to get the optimal particle dynamics and their log-weights evolve as

$$\frac{d}{dt} x(t) = v_{\text{total}}(t, x) = v_{\text{prior}}(t, x) + \nabla_x s(t, x), \qquad \frac{d}{dt} \log w_t(x(t)) = s(t, x(t)).$$

**Corollary 3.3** (HJB residual objective). *Motivated by the duality form, we propose to parameterize* $s_\theta(t, x)$ *with a neural network and define the* velocity-prior guided HJB residual *as:*

$$r_\theta(t, x) := \partial_t s + \tfrac{1}{2}\|\nabla_x s\|^2 + \nabla_x s \cdot v_{\text{prior}} + \tfrac{1}{2}\, s^2. \tag{9}$$

*Then minimizing*

$$\mathcal{L}_{\text{HJB}}(\theta) = \mathbb{E}_{\rho_0(x)}[s_0(x)] - \mathbb{E}_{\rho_1(x)}[s_1(x)] + \int_0^1 \mathbb{E}_{\rho_t(x)}\big[w(t, x)r_\theta(t, x)^2\big]\, dt \tag{10}$$

*drives* $s_\theta$ *toward dual feasibility. Note that in practice, we use squared residual to prevent positive and negative values from cancellation and add a importance weight* $w(t)$ *trick to reduce variance.*

**Importance Reweighting**  The squared HJB residual can be dominated by a few high-variance outliers (rare cells, sharp local flows), which destabilizes training. To ensure training stability and preventing these extreme high residual outliers, we adopt a simple batch-wise importance reweighting that down-weights large residuals. For a mini-batch $\{(t_i, x_i)\}_{i=1}^B$, let $r_i = \big|r_\theta(t_i, x_{t,i})\big| + \varepsilon$ and with a temperature $\tau > 0$. Then for each sample, the weight is inversely proportions to a temperature-shaped residual as $\tilde{w}_i \propto r_i^{-\tau}$. Thus, larger residuals get smaller weight, which reduces variance while keeping the update focused and stable.

**Theorem 3.4** (Prior-guided HJB optimality). *Suppose that the HBJ residual defined in corollary.3.3 satisfies* $r_\theta(t, x) = 0$ *for* $\rho_t$-*a.e on* $[0, 1] \times \mathbb{R}^d$, *and the boundary constraints hold, then* $(\rho_t, v_{\text{corr}}, g_\theta)$ *satisfies the unbalanced continuity equation and the WFR optimality conditions in Definition 3.2. In particular, the learned corrective field* $v_{\text{corr}}^* = \nabla_x s_\theta$ *and growth* $g^* = g_\theta$ *satisfy the optimality conditions. (See Appendix B for proof).*

While the HJB residual enforces local optimality conditions, it does not guarantee that the terminal distribution $\rho_1^\theta$ matches with the target $\rho_1$. To bridge this gap, we design a two-part reconstruction objective: (i) a density matching term through the sliced Wasserstein between the predicted $\hat{\rho}_1$ and $\rho_1$, and (ii) a mass term aligning the global log-mass ratio. These two terms directly calibrates the terminal distribution's *shape* and *mass*, complementing the HJB residual.

**Reconstruction loss**  To align the terminal distribution in *shape*, we use a sliced Wasserstein objective. Let $\hat{\rho}_1$ be the empirical terminal distribution learned from our model, and $\rho_1$ be the ground truth target distribution. $\theta_\ell$ is a random projection sampled from $\theta_\ell \sim \text{Unif}(\mathbb{S}^{d-1})$ for $\ell = 1, \ldots, L$, $\hat{X}_1$ are predicted samples, and $Y$ are ground-truth samples, the sliced Wasserstein loss is defined as:

$$\text{SW}_2^2(\hat{\rho}_1, \rho_1) \approx \frac{1}{L}\sum_{\ell=1}^L W_2^2\Big(\langle \theta_\ell, \hat{X}_1\rangle,\ \langle \theta_\ell, Y\rangle\Big), \tag{11}$$

where $W_2^2$ on $\mathbb{R}$ is the 1D Wasserstein distance computed by sorting projections.

To capture *global mass change*, we track the evolution of particle weights along the learned dynamics. Assuming a WFR scale of $\delta = 1$ and a mini-batch of size $B$, we initialize log-weights as $\log w_i(0) = 0$. The weights evolve according to the potential $s_\theta$ through $\frac{d}{dt}\log w_i(t) = s_\theta\big(t, x_i(t)\big)$, which yields terminal log-weights $\log w_i(1)$. At $t = 0$ the total mass is $M(0) = \sum_{i=1}^B w_i(0) = B$ while at $t = 1$, it is $M(1) = \sum_{i=1}^B w_i(1) = \sum_{i=1}^B \exp\big(\log w_i(1)\big)$.

To calculate the ground truth mass, we neither have access to the full density $\rho_t$ nor to the absolute scale of $M(T)$. Instead, we use relative mass changes between time points, estimated from the number of observed particles at each time $T_k$, which is the $k$-th time point in a multi-snapshot setting. For instance, on the single cell datasets with interval $[T_k, T_{k+1}]$, let $N_k$ be the number of cells observed at time $T_k$. We can approximate the ground-truth log mass ratio by

$$\log r_{\text{target}, k} = \log \frac{M(T_{k+1})}{M(T_k)} \approx \log\left(\frac{N_{k+1}}{N_k}\right), \tag{12}$$

The model's predicted ratios $\log \hat{r}_{\text{model}, k}$ is obtained from the weight evolution described above. We then penalize deviations from the ground truth ratio $r$ in the logarithmic form to ensure stability and

to enforce WFR consistency: $\mathcal{L}_{\text{mass}} = \big( \log \hat{r} - \log r \big)^2$. Our final reconstruction loss combines the two components with tunable coefficients:

$$\mathcal{L}_{\text{recon}} = \lambda_{\text{sw}} \, \text{SW}_2^2(\hat{\rho}_1, \rho_1) + \lambda_{\text{mass}} \big( \log \hat{r} - \log r \big)^2 \tag{13}$$

with $\lambda_{\text{sw}} > 0$ and $\lambda_{\text{mass}} > 0$. In practice we use $L \in [64, 512]$ random projections, and choose $\lambda_{\text{mass}} \in [0.1, 1]$ to calibrate mass without overpowering other terms.

**Total objective**    Putting the pieces together, our total training loss is

$$\min_{\theta} \, \mathcal{L}(\theta) = \underbrace{\mathbb{E}_{x \sim \rho_0}[s_0(x)] - \mathbb{E}_{x \sim \rho_1}[s_1(x)] + \lambda_{\text{hjb}} \int_0^1 \mathbb{E}_{x \sim \rho_t}\big[w(t,x) \, r_\theta(t,x)^2\big] \, dt}_{\mathcal{L}_{\text{HJB}}} \tag{14}$$

$$+ \underbrace{\lambda_{\text{sw}} \, \text{SW}_2^2(\hat{\rho}_1, \rho_1) + \lambda_{\text{mass}} \big( \log \hat{r} - \log r \big)^2}_{\mathcal{L}_{\text{recon}}}$$

where $r_\theta(t,x)$ is the HJB residual, $w(t,x)$ are the nonnegative weights, and $\hat{\rho}_1(\theta)$ is the terminal distribution obtained by pushing $\rho_0$ through the learned dynamics.

## 4 RELATED WORKS

**Physics-constrained approaches**    Existing works (Koshizuka and Sato, 2022; Neklyudov et al., 2023b; Tong et al., 2020) add a *potential-based prior* $V_t(x)$ to the HJ equation to incorporate prior knowledge for trajectory inference. In (Neklyudov et al., 2023b), this yields $\partial_t s + \frac{1}{2}\|\nabla s\|^2 + V_t + \frac{1}{2\delta^2} s^2 = 0$ and the conservative second–order law $\ddot{X}_t = -\nabla V_t(X_t)$. Meanwhile, our method is the *velocity-based prior* approach, the known velocity field $v_{\text{prior}}$ enters as a drift inside the continuity equation, which produces the cross-term $\nabla s \cdot v_{\text{prior}}$. Since potential-based priors is curl-free, it can not represent rotational flows as in (Neklyudov et al., 2023a); By contrast, our data-driven formulation uses a flexible measured vector fields directly as $v_{\text{prior}}$, which can be considered as a "free drift" and learns only minimal optimal corrections and growth.

Most recently, Curly FM (Petrović et al., 2025) proposes a two-stage pipeline: it first learns a smooth global velocity field from approximate velocities (e.g., RNA velocity), and then solves a Schrödinger bridge problem with this learned nonzero drift. In contrast, our VP–HJF framework does *not* require an explicit first-stage training step to learn a reference drift. We directly incorporate the velocity prior as the drift in a Hamilton–Jacobi dual formulation. Moreover, the dynamic prior in Gu et al. (2024) is assumed to be a clean, well-specified prior. Our setting is more flexible by using the corrective field $v_{\text{corr}}$ to adjust noisy or misspecified priors.

**Other trajectory inference approaches**    Trajectory inference has also advanced through flow matching(Haviv et al., 2024; Kapusniak et al., 2024; Atanackovic et al., 2024; Eyring et al., 2023), and Schrödinger bridge methods, which scale effectively to high-dimensional data. Recent SB variants further improve performance on single-cell datasets include (Huguet et al., 2022; Tong et al., 2023b; Shen et al., 2024; Hong et al., 2025; Pariset et al., 2023; Lavenant et al., 2024). For unbalanced settings, variational and regularized UOT methods such as TIGON, DeepRUOT, Var-RUOT, VGFM,and UMFSB directly learn transport dynamics and growth from snapshot data (Sha et al., 2024; Sun et al., 2025; Wang et al., 2025; Zhang et al., 2025). In particular, Var-RUOT also uses a single network to model both the velocity field and the growth term, but it did not incorporate prior known knowledge like ours. Furthermore,Var-RUOT trains a global-in-time trajectory and evaluates its objective by integrating the dynamics over the entire time horizon, which requires simulating the full trajectory on a fine time grid. In contrast, we use a local per interval training scheme combined with a mixture-based sampling strategy to interpolate data between each interval.

---

**Algorithm 1** Training VP–HJF (Velocity-Prior Hamiltonian–Jacobi Flow)

---

**Require:** per time-interval $t_k$ snapshots $\{x, t_k, v_k^{\text{prior}}\}_{k=0}^K$, network $s_\theta(t, x)$, coefficients $\alpha, \beta, \lambda_{\text{SW}}, \lambda_{\text{mass}}$, batch size B

1: **while** not converged **do**
2:      Sample a global batch across all intervals $(x, t, v)_{b=1}^B \sim \{x, t, v_{\text{prior}}\}_{k=0}^K$
3:      Compute $s_0 \leftarrow s_\theta(t_0, x_0), s_1 \leftarrow s_\theta(t_1, x_1)$
4:      **for** $k = 0$ to $K - 1$ **do**
5:          Sample adjacent pairs $(x_k, v_k, x_{k+1}, v_{k+1})_{b=1}^B \sim \{x, t, v_{\text{prior}}\}_{k=0}^K$
6:          **HJB residual loss for interval** $[t_k, t_{k+1}]$**:**
7:          **for** $m = 1$ to $M_{\text{HJB}}$ **do**
8:              Sample normalized $u^{(m)} \sim \mathcal{U}(0, 1)$
9:              Sample $x_{t,k}$ from the mixture $\rho_{t,k}$ of $\rho_k$ and $\rho_{k+1}$, for $b = 1, \dots, B$
10:            Compute $r_{t,k} \leftarrow \partial_t s_{t,k} + \frac{1}{2}\|\nabla_x s_{t,k}\|^2 + \nabla_x s_{t,k} \cdot v_{\text{prior}}\left(t_k^{(m)}, x_{t,k}\right) + \frac{1}{2}\left(s_{t,k}\right)^2$
11:            Compute importance weights $\tilde{w}_b \propto \left|r_{t,k}^{(b)}\right|^{-\tau}$
12:            $\mathcal{L}_{\text{HJB},k} \leftarrow \mathcal{L}_{\text{HJB},k} + \frac{1}{B}\sum_{b=1}^B w_b \left(r_{t,k}\right)^2$
13:          **end for**
14:          **Reconstruction loss:**
15:          Define ODE rhs: x      $\dot{x} = \nabla_x s_\theta(t, x) + v_{\text{prior}}(t, x), \quad \dot{\log w} = s_\theta(t, x)$
16:          Compute $(x_{k+1}^{\text{pred}}, \log w_{k+1}) \leftarrow \texttt{odeint}\Big(\text{ode rhs}, (x_k, \mathbf{0}), \ t \in [t_k, t_{k+1}]\Big)$
17:          $\mathcal{L}_{\text{recon}} \leftarrow \mathcal{L}_{\text{recon}} + \lambda_{\text{SW}} \frac{1}{L}\sum_{\ell=1}^L W_2^2\left(x_{k+1}^{\text{pred}}, x_{k+1}\right) + \lambda_{\text{mass}}(\log \hat{r}_k - \log r_k^2)$
18:      **end for**
19:      **Total loss:**    $\mathcal{L}(\theta) \leftarrow \alpha\,\mathcal{L}_{\text{HJB}} + \beta\,\mathcal{L}_{\text{recon}}$
20:      Update $\theta \leftarrow \theta - \eta\,\nabla_\theta \mathcal{L}(\theta)$
21: **end while**

---

# 5 EXPERIMENTS

## 5.1 SYNTHETIC DATASET

**Balanced case - Rotating Ring** First, we show a case when utilizing the velocity prior is crucial in learning the correct velocity field where curl-free methods like AM fails. We tested on a 2D rotating ring dataset where the points on the ring (source) are rotated by a fixed angle $\theta$ (target). The velocity prior is defined as $v_{\text{prior}} = \omega J x$, where J is the skew-symmetric rotation matrix and $x \in \mathbb{R}^2$, so this becomes $Jx = \begin{bmatrix} 0 & 1; & -1 & 0 \end{bmatrix}\begin{bmatrix} x_1 & x_2 \end{bmatrix}^\top = (-x_2, x_1)^\top$. The task for our model $v_{\text{ot}}$ is to learn the residual correction after given the prior rotation knowledge, such as ensuring the boundary condition by aligning the mismatched source and target density or correcting the radial drift by push the points inwards or outwards,etc. In Figure 1 left (b), we show that since AM has the curl-free limitation, without a prior, its model $\nabla s_t(x)$ failed to represent pure rotation where the streamlines cut through the circle. In Figure 1 left (a), the streamlines from our method form a circular flow indicating that $v_{\text{prior}}$ gives the model the correct inductive bias.

**Diverging Petal** We created a curved and rotated petal-shape dataset to test our method on diverging multi-trajectory paths. The source is a gaussian distribution concentrated in the center and $v_{\text{prior}=\omega J x}$ the rotation dynamic defined as before. Our task is to learn $v_{\text{petal}}$, a radial and angle-dependent term that push outward the points along the radius with different speed depend on the angle $\theta = \text{atan2}(y, x)$. We have

$$v_{\text{petal}}(x) \;=\; s(\theta)\,\hat{r}, \quad \hat{r} = \frac{x}{\|x\|}, \quad s(\theta) = \max\big(0, \; b + a\cos(k\theta)\big) \tag{15}$$

Compared with the petal shape appeared in AM and MIOFlow (Huguet et al., 2022), the underlying dynamic flow in our example is harder to learn, where the former one has a straight-axis aligned radial expansion as $r(x) = |x_1| + |x_2|$ with the gradient of $r(x)$ being a piece-wise constant and curl-free. Figure 1 middle shows that our petal shape matches with the target shape. Figure 1 right shows that the vector field (red arrows) are bending away from pure rotation (blue arrows) to align strongly with the petal shape by pushing the mass outward.

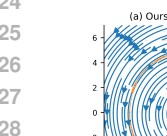 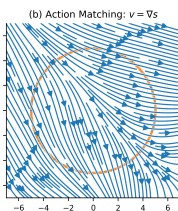 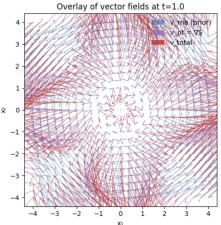 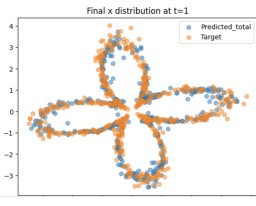

Figure 1: **Left** (first two): Ours correctly learned the rotating dynamic while AM failed. **Middle**: Vector field (red) bending away from $v_{\text{prior}}$ (blue) to form petal shapes. **Right**: Predicted target distribution matches with ground truth

Table 2: Effect of adding $\mathcal{L}_{\text{mass}}$ for Lokta-Volterra. $r_{\text{pred}}$ is the predicted mass ratio , the absolute log error $|\Delta \log r| = |\log r_{\text{pred}} - \log r_{\text{true}}|$, the relative error with $r_{\text{true}} = 1.418$ and the velocity RMSE.

| **Method** | $r_{\text{pred}}$ | $|\Delta \log r|$ | Rel. % err. | Vel. RMSE |
|---|---|---|---|---|
| VP–HJF (ours) | 1.356 | 0.044 | 4.30% | 0.137 |
| VP–HJF (w/o $\mathcal{L}_{\text{mass}}$) | 0.899 | 0.455 | 36.56% | 0.158 |
| Unbalanced AM | 0.938 | 0.413 | 33.80% | 0.302 |
| Prior-only | 1.000 | 0.348 | 29.40% | 0.050 |

**Balanced case - Gaussian Translation**  In this experiment, we compare the least-action or energy costs across different methods. We define an affine prior drift as $v_{\text{prior}}(x) = \mu_1 - \mu_0$ with the parameters $\mu_0 = (0, 0)$, $\mu_1 = (0.5, 6.0)$, $\Sigma_0 = ((2.5, 0.0), (0.0, 0.3))$, $\Sigma_1 = ((0.4, 0.0), (0.0, 2.2))$. As shown in Table 1, our proposed VP-HJF achieves Wasserstein distance $W_2$ accuracy on par with FM Lipman et al. (2022) while OT-FM Tong et al. (2023a); Pooladian et al. (2023) attains the lowest Wasserstein distance. By contrast, both FM and OT-FM must learn the entire velocity field $u_t = \nabla s + w_t$, resulting in a much larger control action. This demonstrates that VP-HJF leverages the structured prior effectively, where the prior dynamics carry most of the transport, and the learned correction $\nabla s$ makes adjustments. To verify that our improvement is not solely due to a strong velocity prior $v_{\text{prior}}$ itself, we also report a prior-only baseline. The prior alone shows moderate accuracy but with high kinetic cost, whereas our method balances both accuracy and energy efficiency.

**Lotka–Volterra with growth.**  We model the prey and predator densities $x_1(t), x_2(t)$ by a first–order nonlinear ODE, $\dot{x}_1(t) = \alpha\, x_1(t) - \beta\, x_1(t)\, x_2(t)$, $\dot{x}_2(t) = -\gamma\, x_2(t) + \delta\, x_1(t)\, x_2(t)$, where $\alpha$ is the prey's intrinsic growth rate, $\beta$ is the predation rate, $\gamma$ is the predator's mortality rate, and $\delta$ is the predator's growth rate from consuming prey(Goel et al., 1971). To model the population expansion and decay dynamics, we use a simple scalar growth field and evolve local mass via the weight dynamics $g(x(t)) = \kappa \left( x_1(t) - x_2(t) \right)$, $\frac{d}{dt} \log w(t) = g(x(t))$ The total mass $M(t) = \mathbb{E}[w(t)]$ and the ground-truth mass ratio is calculated as $r_{\text{true}} = M(t)/M(0)$. We define the $v_{\text{prior}}$ asprior e oracle LV drift with added Gaussian noise with no growth term.

Table 2 shows that adding the explicit mass term $\mathcal{L}_{\text{mass}}$ enables our method to closely match $r_{\text{true}}$ with $4\%$ of relative error while both unbalanced AM and our method without the $\mathcal{L}_{mass}$ term suffer from a much higher relative error of over $30\%$. Although AM aligns transport but leaves the *scale* of the scalar potential $s_\theta$ unconstrained, so the integrated growth $\int_0^t g(x(t))\, dt$ is miscalibrated. By contrast, $\mathcal{L}_{\text{mass}}$ provides endpoint constraint on the mass – yielding better mass dynamics and more aligned with the ground truth mass ratio.

## 5.2  REAL-WORLD DATASET

In this section, we evaluate our method on two single-cell RNA-seq datasets. Both provide RNA velocity, which we use as the velocity prior $v_{\text{prior}}$. Such priors are common in biological and scientific applications beyond single-cell data. Incorporating them introduces an inductive bias that reduces

Table 3: Comparison on the EB dataset using SWD, MMD and W1, at the held-out marginals ($t_1$, $t_3$). Baseline results other than * are taken from (Theodoropoulos et al., 2025)

| Method | SWD $t_1$ | SWD $t_3$ | MMD $t_1$ | MMD $t_3$ | $W_1 t_1$ | $W_1 t_3$ |
|---|---|---|---|---|---|---|
| DeepRUOT | 0.73 | 0.67 | 0.43 | 0.36 | 13.45 | 14.90 |
| Var-RUOT* | **0.37** | **0.24** | 0.25 | **0.06** | **10.28** | **11.92** |
| MIOFlow | 0.84 | 0.94 | 1.01 | 0.92 | 13.20 | 13.57 |
| SBIRR | 0.80 | 0.91 | 0.71 | 0.73 | 15.09 | 20.39 |
| MMFM | 0.59 | 0.76 | 0.37 | 0.35 | 13.61 | 14.64 |
| DMSB | 0.58 | 0.54 | 0.38 | 0.36 | 14.08 | 15.22 |
| 3MSBM | 0.48 | 0.38 | **0.14** | 0.18 | 13.89 | 13.11 |
| **VP-HJF (ours)*** | **0.37** | 0.47 | 0.18 | 0.17 | 11.83 | 13.98 |

Table 4: Robustness analysis of VP–HJF to perturbations of the velocity prior on the EB dataset of 100 dim. We report mean $\pm$ std over 5 seeds

| | Clean | Gaussian noise $\eta$ | | scale $c$ | |
|---|---|---|---|---|---|
| | | 0.25 | 0.75 | 0.5 | 1.5 |
| $W_1 \, t_2$ | 13.26$\pm$ 0.08 | 13.19$\pm$ 0.08 | 13.19 $\pm$0.08 | 13.22 $\pm$0.08 | 13.21 $\pm$ 0.08 |
| $W_1 \, t_4$ | 14.59$\pm$0.09 | 14.58$\pm$0.10 | 14.57 $\pm$0.09 | 14.56$\pm$ 0.10 | 14.62 $\pm$ 0.09 |

learning complexity — our model needs only to learn a corrective flow and growth rather than the full dynamics from scratch.

**EB scRNA-Seq data**   We evaluate cell-trajectory inference on the Embryoid Body (EB) dataset of Moon et al. (2019), using the preprocessed release from Koshizuka and Sato (2022); Tong et al. (2020). The dataset comprises five snapshots over 27 days, grouped as $t_0 \in [0,3]$, $t_1 \in [6,9]$, $t_2 \in [12,15]$, $t_3 \in [18,21]$, $t_4 \in [24,27]$. Leveraging RNA velocity as a prior $v_{\text{prior}}$ at each snapshot, we train a local and shorter trajectory by adopting the multi-marginal *local per-interval* training: at each step we sample an adjacent pair $(t_k, t_{k+1})$ and learn only the transport and growth to move $\rho_{t_k} \to \rho_{t_{k+1}}$. This yields more stable gradients and low target variance than enforcing all time points jointly. For details of the training algorithm, see Algorithm 1.

We test on 100-dim PCA components feature space and compare with recent works using the multi-marginal approach from 3MSBM (Theodoropoulos et al., 2025), SBIRR (Shen et al., 2024), MMFM (Rohbeck et al., 2025) and DMSB (Chen et al., 2023) as well as other methods using global-in-time joint training or unbalanced optimal transport DeepRUOT (Zhang et al., 2024), Var-RUOT (Sun et al., 2025) and MIOFlow (Huguet et al., 2022). We follow the experiment setup from 3MSBM by having $t = 1, 3$ as the held-out sets and evaluating on various metrics. Table 3 shows that our method outperforms most methods and remains competitive with Var-RUOT and 3MSBM. Notably, our method outperforms SBIRR and MMFM, which solve piecewise Schrödinger bridges and OT couplings whereas methods like 3MSBM and DMSB solve a single global optimization with a joint coupling, indicating the benefits of using RNA velocity as a local prior with per-interval supervision. For additional results and comparison with Var-RUOT see Appendix C.

We also conducted robustness analysis on the mis-specification of $v_{\text{prior}}$ of our approach. Specifically, we perturb the reference field by (i) adding Gaussian noise, $v_{\text{prior}} = v_{\text{prior}} + \eta$, and (ii) rescaling its magnitude, $v_{\text{prior}} = c \, v_{\text{prior}}$. In Table 4, $\mathcal{W}_1$ at $t_2$ and $t_4$ changes only marginally under both noise levels ($\alpha \in \{0.25, 0.75\}$) and scaling factors ($c \in \{0.5, 1.5\}$). This indicates that VP–HJF is robust under mild to moderate prior perturbations, with the learned corrective field $v_{\text{corr}}$ adapting to and compensating for mis-specification in $v_{\text{prior}}$.

**Bone marrow scRNA-Seq data**   We evaluate our approach on a real scRNA-seq bone-marrow atlas with multiple hematopoietic fates from scVelo (Bergen et al., 2020). Figure 2 (left) shows trajectories that emanates from early progenitor regions at $t{=}0$ (dark blue) and spread out to other branches by closely following the UMAP reference (gray). Figure 2 (right) shows the learned growth

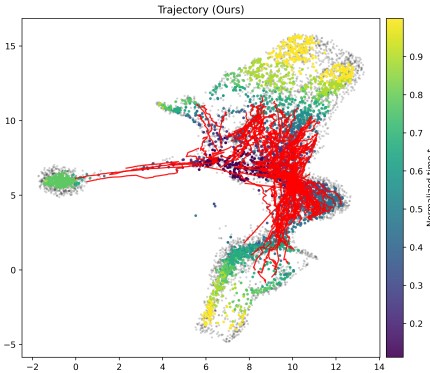 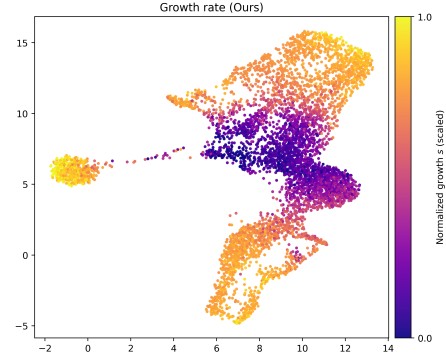

Figure 2: **Left**:Bone marrow trajectories. Colored points (ours) show inferred cell trajectories overlaid on the reference manifold (gray)) **Right**: Learned growth field — orange means high growth, purple means low growth.

field $g_\theta(t, x) = s_\theta(t, x)$ that governs local mass dynamics. This maps shows that our model successfully captured low growth rate in early progenitors and increases as cells enter the active cycling and amplification stage. However, we also observe that the high growth rate near some terminal regions. While this indicates the model can assign higher growth to specific cell types, a strong terminal-phase growth is biologically implausible. For the bone marrow case, mature or exiting cells should have near-zero or negative growth. This likely reflect the objective imbalance where transport terms dominating mass calibration, suggesting mild regularization such as time-smoothness on $s_\theta$ or branch-wise boundary constraints to better align growth with biology.

## 6 DISCUSSION AND LIMITATIONS

**On the velocity prior quality and assumptions** The velocity prior $v_{\mathrm{prior}}$ indeed plays a constructive—but double-edged role in our method. A good prior captures coarse dynamics - reducing the learning complexity and improving on sample efficiency. A mis-specified prior can bias the learned corrective field $s_\theta(t, x)$ and slow or destabilize training. Hence, the *quality* of $v_{\mathrm{prior}}$ strongly influences both optimization and generalization. In practice, mild perturbation of the prior through noise, scale or mis-specification are corrected by $s_\theta$, whereas severe mis-specification such as overly large or structurally wrong drifts can bias the learned corrective flow. Moreover, $v_{\mathrm{prior}}$ does *not* require divergence-free assumption. Our dual objective explicitly includes the cross term $\nabla_x s_\theta \cdot v_{\mathrm{prior}}$ in the HJB residual avoiding hidden orthogonality requirements.

**Limitations** In Fig. 2 (right) we observe high growth near terminal regions, which is biologically implausible for mature or cell–cycle–exiting states. Without additional biological constraints such as cell-cycle markers,branch–terminal boundary conditions or proliferation markers, growth–transport disentanglement may remain under-determined in some regions. A promising direction is to add weak supervision on the learned growth model to improve identifiability.

For single–cell datasets we currently use local supervision—training on adjacent pairs with a time–continuous shared network. This choice is simple and scalable and induces a globally smooth field, but it does not jointly enforce all marginals as in recent multi–marginal methods, which may limit long–range trajectory coherence. In future works, extending VP–HJF with global consistency could improve on long-range trajectory inference.

## 7 CONCLUSION

Our method decomposes the velocity field by using domain knowledge as prior and using a single network to capture both growth and transport. This decomposition yields robust performance even under mild to moderate prior mis-specification, indicating the flexibility of this framework to incorporate priors, making it a promising direction for modeling complex cellular dynamics.

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

# A  PRIOR-GUIDED HJB INEQUALITY DERIVATION

To derive the velocity-prior guided HJB residual inequality in Eq.8.

$$\mathcal{A}(\rho, v, g) \geq \mathbb{E}_{x \sim \rho_0}[s_0(x)] - \mathbb{E}_{x \sim \rho_1}[s_1(x)]$$
$$- \int_0^1 \int \rho_t(x) \left( \partial_t s + \tfrac{1}{2}\|\nabla s\|^2 + \nabla s \cdot v_{\text{prior}} + \tfrac{1}{2}s^2 \right) dx\, dt. \qquad (16)$$

We want to start by minimizing Definition 3.2. Since $v_{\text{prior}}$ is constant and known, we only need to minimize the "learned" action portion $v_{\text{corr}}$, then the problem becomes the following:

Consider the velocity–prior guided WFR action:

$$\min_{\rho,\, v_{\text{ot}},\, g} \quad \mathcal{A}(\rho, v_{\text{corr}}, g) := \int_0^1 \int_\Omega \left( \tfrac{1}{2}\|v_{\text{corr}}(t,x)\|^2 + \tfrac{1}{2}g(t,x)^2 \right) \rho_t(x)\, dx\, dt \qquad (17)$$

$$\text{s.t.} \quad \partial_t \rho_t + \nabla \cdot \big( \rho_t(v_{\text{prior}} + v_{\text{corr}}) \big) = g_t\, \rho_t, \qquad \rho_{|t=0} = \rho_0,\quad \rho_{|t=1} = \rho_1.$$

**Step 1: Lagrangian formulation**
First, we introduce a scalar multiplier $s(t,x)$, the Lagrangian becomes:

$$\mathcal{L} = \int_0^1 \int_\Omega \left( \tfrac{1}{2}\|v_{\text{corr}}\|^2 + \tfrac{1}{2}g^2 \right) \rho\, dx\, dt \qquad (18)$$
$$+ \int_0^1 \int_\Omega s \Big( \partial_t \rho + \nabla \cdot (\rho(v_{\text{prior}} + v_{\text{corr}})) - g\rho \Big) dx\, dt.$$

**Step 2: Integration by parts**
Integration by parts in time on $\int s\partial_t \rho$

$$\int_0^1 \int_\Omega s\, \partial_t \rho\, dx\, dt = \left[ \int_\Omega s\, \rho\, dx \right]_{t=0}^{t=1} - \int_0^1 \int_\Omega \rho\, \partial_t s\, dx\, dt \qquad (19)$$
$$= \mathbb{E}_{x \sim \rho_0}[s_0(x)] - \mathbb{E}_{x \sim \rho_1}[s_1(x)] - \int_0^1 \int_\Omega \rho\, \partial_t s\, dx\, dt.$$

Integration by parts in space on $\int s \nabla \cdot (\rho w)$ with $w := v_{\text{prior}} + v_{\text{ot}}$

$$\int_\Omega s\, \nabla \cdot (\rho w)\, dx = \int_{\partial \Omega} s\, \rho\, w \cdot n\, d\sigma - \int_\Omega \nabla s \cdot (\rho w)\, dx. \qquad (20)$$

Assuming zero boundary flux or fast decay, the boundary term vanishes:

$$\int_0^1 \int_\Omega s\, \nabla \cdot (\rho w)\, dx\, dt = - \int_0^1 \int_\Omega \rho\, w \cdot \nabla s\, dx\, dt. \qquad (21)$$

Then, the Lagrangian becomes

$$\mathcal{L} = \mathbb{E}_{x \sim \rho_0}[s_0(x)] - \mathbb{E}_{x \sim \rho_1}[s_1(x)] \qquad (22)$$
$$+ \int_0^1 \int_\Omega \rho_t(x) \left[ \tfrac{1}{2}\|v_{\text{corr}}\|^2 - v_{\text{corr}} \cdot \nabla s + \tfrac{1}{2}g^2 - s\, g - \partial_t s - \nabla s \cdot v_{\text{prior}} \right] dx\, dt.$$

**Step 3: Fenchel–Young inequality**
The Fenchel–Young inequality states that for any vectors a and p, we have:

$$\tfrac{1}{2}\|a\|^2 \geq p \cdot a - \tfrac{1}{2}\|p\|^2 \qquad (23)$$

We set $a = v_{\text{corr}}$ and $p = \nabla s$, then we have:

$$\tfrac{1}{2}\|v_{\text{corr}}\|^2 - v_{\text{corr}} \cdot \nabla s \geq -\tfrac{1}{2}\|\nabla s\|^2 \qquad (24)$$

Similarly, we set $a = g$ and $p = s$, then we have:

$$\tfrac{1}{2}g^2 - s\, g \geq -\tfrac{1}{2}s^2, \qquad (25)$$

with equality iff $v_{\text{corr}} = \nabla s$ and $g = s$. Thus, putting pieces together, we have:

$$\mathcal{A}(\rho, v_{\text{corr}}, g) \geq \mathbb{E}_{x \sim \rho_0}[s_0(x)] - \mathbb{E}_{x \sim \rho_1}[s_1(x)]$$
$$- \int_0^1 \int_\Omega \rho_t(x) \left( \partial_t s + \tfrac{1}{2}\|\nabla s\|^2 + \nabla s \cdot v_{\text{prior}} + \tfrac{1}{2}s^2 \right) dx\, dt. \qquad (26)$$

# B    PROOF OF THEOREM 3.4

**Theorem B.1** (Prior-guided HJB optimality). *Suppose that the HBJ residual defined in corollary.3.3 satisfies $r_\theta(t, x) = 0$ for $\rho_t$-a.e on $[0, 1] \times \mathbb{R}^d$, and the boundary constraints hold, then $(\rho_t, v_{\mathrm{corr}}, g_\theta)$ satisfies the unbalanced continuity equation and the WFR optimality conditions in Definition 3.2. In particular, the learned corrective field $v^*_{\mathrm{corr}} = \nabla_x s_\theta$ and growth $g^* = g_\theta$ satisfy the optimality conditions.*

**Proof sketch**    The proof of this theorem builds upon the derivations from the previous proof. Recall, from step 3 Fenchel–Young inequality above, we have:

$$\mathcal{A}(\rho, v_{\mathrm{corr}}, g) \geq \mathbb{E}_{x \sim \rho_0}[s_0(x)] - \mathbb{E}_{x \sim \rho_1}[s_1(x)] - \int_0^1 \int_\Omega \rho_t(x)\, r_s(t, x)\, dx\, dt. \tag{27}$$

where the HJB residual is defined as:

$$r_s(t, x) := \partial_t s + \tfrac{1}{2}\|\nabla_x s\|^2 + \nabla_x s \cdot v_{\mathrm{prior}} + \tfrac{1}{2}s^2. \tag{28}$$

**Step 1 (Dual constraint and lower bound)**
To ensure a finite dual lower bound, we need to restrict to potentials $s$ satisfying:

$$r_s(t, x) \geq 0 \quad \forall (t, x). \tag{29}$$

Then under this constraint, since the last term is nonpositive, we have:

$$\mathcal{A}(\rho, v_{\mathrm{corr}}, g) \geq \mathbb{E}_{x \sim \rho_0}[s_0(x)] - \mathbb{E}_{x \sim \rho_1}[s_1(x)] \quad \forall s \tag{30}$$

This inequality also holds when we take the infimum over all feasible $(\rho, v_{\mathrm{corr}}, g)$, which gives the dual problem

$$\inf_{\rho, v_{\mathrm{corr}}, g} \mathcal{A}(\rho, v_{\mathrm{corr}}, g) \geq \sup_{s\,:\,r_s \geq 0} \left( \mathbb{E}_{x \sim \rho_0}[s_0(x)] - \mathbb{E}_{x \sim \rho_1}[s_1(x)] \right). \tag{31}$$

**Step 2 (Residual Optimality)**
Now suppose there exists a potential $s_\theta$ and a feasible triplet $(\rho_t, v_{\mathrm{corr}}, g)$ such that

$$r_{s_\theta}(t, x) = 0 \quad \text{for } \rho_t\text{-a.e.,} \tag{32}$$

$$v_{\mathrm{corr}} = \nabla_x s_\theta, \quad g = s_\theta, \tag{33}$$

and the boundary constraints $\rho_{|t=0} = \rho_0$, $\rho_{|t=1} = \rho_1$ hold.

Then the Fenchel–Young inequalities are equalities and equation 27 becomes:

$$\mathcal{A}(\rho, v_{\mathrm{corr}}, g) = \mathbb{E}_{x \sim \rho_0}[s_\theta(0, x)] - \mathbb{E}_{x \sim \rho_1}[s_\theta(1, x)] - \int_0^1 \int_\Omega \rho_t(x)\, r_{s_\theta}(t, x)\, dx\, dt. \tag{34}$$

Since $r_{s_\theta} = 0$ $\rho$-a.e., then we have:

$$\mathcal{A}(\rho, v_{\mathrm{corr}}, g) = \mathbb{E}_{x \sim \rho_0}[s_\theta(0, x)] - \mathbb{E}_{x \sim \rho_1}[s_\theta(1, x)]. \tag{35}$$

Combining this with the dual lower bound, we have:

$$\inf_{\rho, v_{\mathrm{corr}}, g} \mathcal{A}(\rho, v_{\mathrm{corr}}, g) < \mathcal{A}(\rho, v_{\mathrm{corr}}, g) \tag{36}$$

$$\mathbb{E}_{x \sim \rho_0}[s_\theta(0, x)] - \mathbb{E}_{x \sim \rho_1}[s_\theta(1, x)] < \sup_{s\,:\,r_s \geq 0} \left( \mathbb{E}_{x \sim \rho_0}[s_0(x)] - \mathbb{E}_{x \sim \rho_1}[s_1(x)] \right). \tag{37}$$

So we have:

$$\mathcal{A}(\rho, v_{\mathrm{corr}}, g) = \inf_{\rho, v_{\mathrm{corr}}, g} \mathcal{A}(\rho', v'_{\mathrm{corr}}, g') = \sup_{s\,:\,r_s \geq 0} \left( \mathbb{E}_{x \sim \rho_0}[s_0(x)] - \mathbb{E}_{x \sim \rho_1}[s_1(x)] \right), \tag{38}$$

Finally, we conclude that $(\rho_t, v_{\mathrm{corr}}, g)$ is primal optimal. In particular,

$$v^*_{\mathrm{corr}} = \nabla_x s_\theta, \qquad g^* = s_\theta,$$

and $(\rho_t, v^*_{\mathrm{corr}}, g^*)$ satisfies the unbalanced continuity equation and the WFR optimality conditions in Definition 3.2.

Table 5: Comparison on the EB dataset using $W_2$ at the held-out marginals $(t_1, t_3)$. Baseline results other than * are taken from (Theodoropoulos et al., 2025).

| Method | $W_2\ t_1$ | $W_2\ t_3$ |
|---|---|---|
| DeepRUOT | 13.64 | 15.10 |
| Var-RUOT* | 10.34 | 12.02 |
| MIOFlow | 13.66 | 14.05 |
| SBIRR | 15.42 | 20.98 |
| MMFM | 14.68 | 14.83 |
| DMSB | 14.83 | 15.49 |
| 3MSBM | 14.51 | 13.26 |
| **VP-HJF (ours)*** | 11.94 | 12.28 |

Table 6: Comparison on the EB dataset using weighted $W_1$ at marginals $t_1$–$t_4$ for Var-RUOT and VP-HJF.

| Method | $W_1^{\text{weighted}}$ | | | |
|---|---|---|---|---|
| | $t_1$ | $t_2$ | $t_3$ | $t_4$ |
| Var-RUOT* | 10.28 | 11.58 | 11.90 | 13.28 |
| **VP-HJF (ours)*** | 11.14 | 12.45 | 13.14 | 14.45 |

## C  ADDITIONAL EXPERIMENTS

We conducted further comparison on the EB dataset with 100-dim using the $\mathcal{W}_2$ metric at the held-out marginals at $t_1, t_3$. Table 5 shows that our approach remains competitive and outperforms most methods. We then compare more directly with VAR-RUOT in Table 6 using the weighted $\mathcal{W}_1$, since both methods are in the unbalanced optimal transport setting. In this evaluation, we assign non-uniform particle weights by integrating the learned dynamics through ODE integration and use these predicted weights when computing $\mathcal{W}_1$, instead of uniform masses.

On this weighted $\mathcal{W}_1$ metric, Var-RUOT achieves slightly lower values than VP–HJF. This gap could partly due to the use of noisy RNA-velocity as $v_{\text{prior}}$ in our framework, which can trade a small increase in transport cost for better agreement with the measured dynamics. In addition, Var-RUOT optimizes a single global-in-time trajectory via SDE simulations, whereas VP–HJF relies on deterministic ODE rollouts with local per-interval supervision.

**further training details on single-cell datasets**  For both the EB and bone marrow datasets, we use a 4-layer MLP with Swish activation. The MLP outputs follows the Action Matching implementation, where the output $h_\theta(t, x)$ needs to multiply by the original data input x so the scaler output becomes $s = (h \times x).sum()$ Moreover, We optimize the model with Adam and set the learning rate to $2e - 4$ for both datasets. We follow Algorithm 1 to train with 300 epochs and 256 batch size, using dopri5 with 16 steps for the ode integration. We set the HBJ loss coefficients to $\lambda_{\text{hbj}} = 0.01$ , sliced Wasserstein loss coefficient $\lambda_{\text{sw}} = 10$ and the mass loss coefficient to $\lambda_{\text{mass}} = 0.01$.

For the Var-RUOT baseline on EB, we use the authors' publicly released implementation and configuration, changing only the training epoch to 500 epochs.

## D  LLM USAGE

We used LLM for improve on writing, mainly for checking grammar. We also used LLM for finding relevant and related works.

