# OpenReview forum: "Learning Velocity Prior-Guided Hamiltonian-Jacobi Flows with Unbalanced Optimal Transport"
_ICLR.cc/2026/Conference — Submitted to ICLR 2026_

### Official Review · Reviewer_r9zW · 2025-10-27

**Soundness:** 2
**Presentation:** 3
**Contribution:** 2
**Rating:** 2
**Confidence:** 5

**Summary:**

This paper introduces a method called VP-HJF that incorporates a velocity prior into Hamilton– Jacobi–Bellman flows to reconstruct single-cell trajectories. The approach also extends to unbalanced optimal transport by including a growth term and trains a single neural network to model the corrective velocity. The main claimed contributions are: (i) leveraging a velocity prior to capture complex transport patterns including rotational flows, (ii) introducing a growth term for unbalanced transport, and (iii) reducing learning complexity by only modeling the corrective velocity rather than the full dynamics.

**Strengths:**

1. The paper introduces an intuitive idea of incorporating a known or measurable velocity prior into the unbalanced OT / WFR formulation, resulting in the proposed Velocity-Prior Hamilton–Jacobi Flow (VP-HJF). The decomposition is natural and helps separate coarse drift from residual correction, potentially improving the results.
2. The paper is well written and structured.

**Weaknesses:**

1.	The main weakness lies in the incremental nature of the contribution. Incorporating a velocity prior to improve trajectory inference is not entirely new; see, for example, (Gu et al., ICLR 2025), which also introduces prior-guided dynamics. Also, a very relevant reference is Curl flow matching (Petrović et al., NeurIPS 2025).  I think the author missed these important references. In addition, using a single network to solve (unbalanced) OT or RUOT (regularized unbalanced optimal transport)-style problems has already been demonstrated in (Neklyudov et al., ICML 2024; Sun et al., NeurIPS 2025), which derives and employs the general HJB formulation. Combining these two ideas naturally leads to the present method, making the contribution incremental in my view.
2. In Figure 2 (upper table), VP-HJF is reported to achieve lower total kinetic energy than OT-FM. However, its higher $W_2$ distance raises the concern that it might not have correctly mapped $\rho_0$ --> $\rho_1$, which could trivially explain the reduced energy. The authors should rule out this possibility. Moreover, the exact form of the velocity prior $v_{\text{prior}}$ used in this experiment is not clearly specified.
3.  For the synthetic Lotka–Volterra dataset, only mass error is reported, with no analysis of velocity accuracy. The ablation compares only a “Prior-only” baseline but not a “Total velocity (without prior)” counterpart. Again, the form of $v_{\text{prior}}$ is unspecified.
4. The paper claims that VP-HJF excels at modeling rotational flows. However, the biological datasets used—Embryoid Body (EB) and bone marrow—primarily reflect developmental and differentiation trajectories, which are largely monotonic rather than rotational. Demonstrating the method on a cell-cycle dataset with pronounced curl-like behavior (e.g., as in Curl-Flow Matching, NeurIPS 2025) would be more convincing.
5. The synthetic ablations show VP-HJF is sensitive to the orientation of $v_{\text{prior}}$. On real scRNA-seq data, the method uses RNA velocity as the prior, but RNA velocity is known to be noisy and sometimes reversed (Bergen et al., Molecular Systems Biology). This raises questions about robustness. In addition, the paper lacks comparisons to other unbalanced OT–based trajectory inference methods, such as uOT-CFM (Eyring et al., ICLR 2024) and DeepRUOT (Zhang et al., ICLR 2025).
6. In Figure 3, the learned growth rate appears to increase with time. Biologically, stem-like progenitor cells at early timepoints typically exhibit higher proliferation rates, so this trend seems inconsistent. This issue echoes the limitations of the WFR formulation discussed in (Sun et al., NeurIPS 2025), which could be discussed by the authors.

**Questions:**

1. Specify the prior velocity fields. The experiments should clearly describe the functional form or source of each v_{\text{prior}} used—particularly in the Gaussian Translation and Lotka–Volterra experiments.
2. Validate the correctness of the Balanced Gaussian Translation experiment. Since VP-HJF reports lower kinetic energy but higher W_2 distance than OT-FM, could this indicate imperfect matching of $\rho_0$ and $\rho_1$? Please confirm that endpoint constraints are satisfied and that the energy reduction is not an artifact of incomplete transport.
3. Report velocity error in the Lotka–Volterra experiment. Currently, only the mass ratio error is shown. Including velocity RMSE or trajectory error would clarify how accurately the model recovers the dynamic vector field itself.
4. Test on datasets with pronounced rotational dynamics. Since a central claim of the paper is that VP-HJF handles rotational (curl-dominated) flows, please consider evaluating on a biologically relevant cell-cycle dataset or a synthetic curl-flow benchmark similar to Curl-Flow Matching. This would better support the claimed advantage.
5. Discuss robustness to noisy or misaligned priors. The ablations indicate sensitivity to prior direction, and RNA velocity is known to contain substantial noise or reversed directions. Could the authors discuss possible regularization or self-consistency checks to mitigate such issues?
6. Discuss the biological plausibility of the learned growth field. Figure 3 shows high growth at terminal regions, which appears biologically implausible. The authors may wish to discuss this behavior in light of the limitations of the WFR formulation noted by （Sun et al., NeurIPS 2025).

Minor Issues
1. In Equation (6), v should be vcorr to match the decomposition.
2. Lines 305–308: references to Figure 1 middle/right do not match the actual figure panels.
3. Lines 410–411: The claim “This yields more stable gradients and low target variance than enforcing all time points jointly” lacks supporting experiments or citations.
4. Multiple references to “Table 2” in the text actually refer to the tables in Figure 2, which conflicts with the actual Table 2 and causes confusion.

In light of the comments, at the current stage, I lean toward a score of a clear rejection. I believe this work has merit. That said, I am still willing to revise my score if the authors can provide stronger empirical evidence (e.g., the datasets and the baselines), clarify the experimental setups (especially regarding the priors), and further discuss the biological interpretation issues.


References

1. Gu, A., Chien, E., & Greenewald, K. (2024). Partially observed trajectory inference using optimal transport and a dynamics prior. ICLR 2025.
2. Katarina Petrović et al., Curly Flow Matching for Learning Non-gradient Field Dynamics, NeurIPS 2025
3. Sun, Yuhao, et al. "Variational Regularized Unbalanced Optimal Transport: Single Network, Least Action." NeurIPS 2025
4. Kirill Neklyudov et al., "A Computational Framework for Solving Wasserstein Lagrangian Flows", ICML 2024
5. Zhang, Zhenyi et al. "Learning stochastic dynamics from snapshots through regularized unbalanced optimal transport." ICLR 2025.
6. Luca Eyring et al., Unbalancedness in neural monge maps improves unpaired domain translation. ICLR 2024.
7. Volker Bergen, Ruslan A Soldatov, Peter V Kharchenko, and Fabian J Theis. Rna velocity- current challenges and future perspectives. Molecular systems biology, 17(8):e10282, 2021.

The reviewer prepared this evaluation personally. Use of an LLM was limited to minor editorial polishing.

---

> ### Author Response · Authors · 2025-11-19
>
> We thank the review for this very detailed and sincere review to help us make improvements.
>
> 1. **On incremental contribution** - thanks for suggesting (Gu et al., ICLR 2025) and (Petrović et al., NeurIPS 2025) two recent and relevant works that we missed. We will include discussions on these works in the revised version. Briefly, Curly FM has a two-stage training process, which needs to First learning a smooth global velocity field from the approximate velocities, such as the RNA velocity or other known dynamics and then Second solving a Schrödinger-bridge problem with this learned non-zero-drift reference process to match population marginals. In contrast, our VP-HJF framework does not have an explicit first stage training on learning the reference drift. In our approach, we directly incorporates the velocity prior and used it as a free drift inside a Hamilton–Jacobi dual formulation. For the scRNA dataset, instead of using a neural net to learn it, we interpolate between each time-interval given snapshot data for $v_{prior}(t,x)$. Moreover, we focus on a energy-efficient approach by following the least action principle similar in action matching, Regards on (Gu et al., ICLR 2025), our understanding is that its dynamic prior at least used in the experiments (constant velocity) is already a "clean" prior without noise or misspecification. In contrast, since we decompose the velocity into $v_{prior} + v_{corr}$, the corrective part helps to correct noisy priors like RNA velocity, thus it becomes more flexible. Regards on Neklyudov et al., ICML 2024; Sun et al., NeurIPS 2025), we briefly discussed them in the related work section but we will revise it to add more details.
> \\
> 2. **Specify the prior velocity fields** - In the Gaussian translation case, we define an affine prior drift as $  v_{\text{prior}}(x) =\ (\mu_1 - \mu_0) + \eta\(x - \mu_0)$ with the parameters $\mu_0 = (0,0)$, $\mu_1 = (0.5,6.0) $, $\Sigma_0 = ((2.5,0.0),(0.0, 0.3))$, $\Sigma_1 =((0.4, 0.0),(0.0, 2.2)) $ and $\eta=0.0$. We checked end the point matching by comparing the mean and covariance of the pushforward of $\rho_0$ to the target $\rho_1$. VP-HJF has the smallest mean error and the covariance error is comparable to Flow Matching and slightly higher than OT-FM.
>
> Balanced Gaussian experiment with the mean error denote by $\lVert \hat{\mu}_1 - \mu_1 \rVert_2$ and the covariance error by $\lVert \hat{\Sigma}_1 - \Sigma_1 \rVert_F$.
>
> | Method          | Mean error | Covariance error |
> |----------------|-----------:|-----------------:|
> | VP-HJF (ours)  | 0.102      | 0.791            |
> | Flow Matching  | 0.204      | 0.804            |
> | OT-FM          | 0.149      | 0.659            |
>
> In the Lotka–Volterra experiment, $v_{prior}$ is defined as the oracle LV drift with added Gaussian noise and with no growth term, so  $x = (x_1, x_2)^\top$ and $v_{\text{prior}}(x) = (a x_1 - b x_1 x_2,\; - g x_2 + d x_1 x_2)^\top + \varepsilon,$
>
> Below, we expanded Figure 2(bottom) by adding RMSE velocity error and Prior-only baseline:
>
> | Method                 | r_pred | Δ log r (abs) | Rel. % err. | Vel. RMSE |
> |------------------------|:------:|:-------------:|:-----------:|:---------:|
> | VP–HJF (ours)          | 1.356  |    0.044      |   4.30%     |   0.137    |
> | VP–HJF (w/o L_mass)    | 0.899  |    0.455      |  36.56%     |   0.158    |
> | Unbalanced AM          | 0.938  |    0.413      |  33.80%     |   0.302    |
> | Prior-only             | 1.000  |    0.348      |  29.4%      |   0.05    |
>
> Because the prior-only field is almost oracle with true LV drift plus small Gaussian noise, it achieves the lowest velocity RMSE by construction. In contrast, VP-HJF must simultaneously learn the corrective field $v_{corr}$, satisfy the HJB residual, and match the global mass ratio, which slightly increases its RMSE while still keeping it small. Importantly, since LV shows cyclic population dynamics, VP-HJF achieves substantially lower velocity RMSE than unbalanced AM, whose gradient-based parameterization is limited to model periodic dynamics.
>
> 3. **Biological plausibility of the growth model** - we briefly discussed this limitation in Section 6 first paragraph. This growth behavior is not biologically plausible and we will explore ways for future improvements as well.
>
> 4. **Cell-cycle dataset and robustness tests** - thanks for the suggestions, we are working on providing more results regards on this and we will try to provide an update here hopefully in time!
>
> Thanks very much for your time and feedback again!

---

> > ### Comment · Reviewer_r9zW · 2025-11-20
> >
> > I appreciate the authors’ response. The additional clarifications and experiments have addressed part of my concerns, so I am happy to increase my score at this stage. I also look forward to seeing results on more datasets and with additional baselines, and also the robustness tests, which I believe would further strengthen the paper. Therefore, I am still open to adjusting my score further should the authors provide more comprehensive results.

---

### Official Review · Reviewer_Hm4T · 2025-10-31

**Soundness:** 2
**Presentation:** 1
**Contribution:** 2
**Rating:** 2
**Confidence:** 4

**Summary:**

The authors tackle the problem of learning unbalanced population dynamics of physical systems. They propose a novel method, Velocity Prior-Guided Hamiltonian-Jacobi Flows (VP-HJF), which extends unbalanced action matching to use velocity information as a prior on the drift of the system. As a result, VP-HJF is able to improve modeling of unbalanced population dynamics in certain synthetic settings. The authors present a theoretical backing for the framework and evaluate their approach on a real data setting (modeling cell dynamics using single-cell RNA-seq data).

**Strengths:**

- In general, the authors present novel and principled method useful for leveraging prior "velocity" knowledge for guiding/learning unbalanced optimal transport dynamics.
- The work is well motivated with many potential downstream applications for modeling population dynamics in physical systems.

**Weaknesses:**

I believe this work presents an interesting direction of research and has the potential to provide a useful contribution to the respective community. With that being said, there are several elements that I think need to be addressed to strengthen this work and provide fairer evaluation. Please see below.

- The proposed approach, VP-HJF, seems to require numerical simulation during training to compute the additional density matching loss-term, compared to counterpart methods which are simulation-free during training.
- Empirical experiments could benefit from additional attention, i.e. more datasets/applications/settings. For example, see [2, 4, 5] for some examples of applicable datasets related to the setting(s) considered in this work.
- Moreover, some baselines which are applied to the same problem settings (especially in the scRNA-seq dataset) are missing. Please see [1, 2, 3, 4, 5]. In addition, unbalanced action matching should also be considered as a baseline for the scRNA-seq experiment. To add, it does not appear the proposed method achieves the best performance on the scRNA-seq dataset, weakening the main claims of this work.
- In regard to what the authors label as a "velocity-based prior" approach, [5] directly addresses what appears to be a very similar problem, but with a simulation-free approach. Granted, they do not consider the unbalanced case, but nonetheless should be considered as a baseline as a velocity-based prior approach.
- There is no experiment comparing the computational cost and efficiency of VP-HJF with existing baselines and should be included (especially considering the proposed method is not simulation-free during training).
- The paper at times feels unfinished. For example, the conclusion is only 1 sentence, the introduction is missing clear statements of contributions, the references are limited (apart from those I directly mentioned here, there are more related works pertinent to this field which need to be cited), and the appendix seems incomplete: i.e. additional details for experiments, models training and hyper-parameters, training time and compute usage, dataset information, etc ... are missing.

**Questions:**

- My questions are primarily oriented with the items I listed above under weaknesses, thus, I have no explicit questions at this time.

Minor comments:

- No references to locations for the proofs of Corollary 3.3 and Theorem 3.4.
- Lines 193-194: $\text{Unif}(\mathbb{S}^{d - 1})$ is introduced but not defined.

References:

[1] Kapusniak et al. "Metric flow matching for smooth interpolations on the data manifold." NeurIPS. 2024.

[2] Neklyudov et al. "A computational framework for solving wasserstein lagrangian flows." ICML. 2024.

[3] Wang et al. "Joint Velocity-Growth Flow Matching for Single-Cell Dynamics Modeling." arXiv. 2025.

[4] Zhang et al. "Modeling Cell Dynamics and Interactions with Unbalanced Mean Field Schrodinger Bridge." arXiv. 2025

[5] Petrović et al. "Curly flow matching for learning non-gradient field dynamics." ICLR 2025 Workshop on Machine Learning for Genomics Explorations. 2025.

---

> ### Author Response · Authors · 2025-12-03
>
> Thank you for the detailed reveal and suggestions !
>
> **On additional baselines** In our revised version, we have conducted further evaluations on the EB 100-dim dataset using SWD, MMD, $w_1$ and $w_2$, and we also added new baselines including DeepRUOT, Var-RUOT (these two also use the unbalanced OT setting) and MIOFlow. Please see table 3,4,5,6.  Regards on additional experiments, thanks for listing potential suggestions from other related work. we will include additional results in the future.
>
> **On comparison with related work** In our revised version, we included a comparison with with Curly FM. Briefly, Curly FM uses a two-stage training which needs to learn a reference drift using RNA velocity first. In comparison, our formulation incorporates it directly without additional training and use it as a "free drift". For future version, we will include additional experiment using relevant  dataset such as the cell cycle dataset from this paper and conduct further comparison. Given the time constraints, we were not able to include the result in this version of the revised draft.
>
> **Additional clarification** In our revised version, we included the proof for Theorem 3.4. Thanks for your honest suggestion on the last bullet point. We also have expanded the conclusion section, added more references and provided training details on the single cell datasets in the appendix. Regards on simulation-free, as you have pointed our, our method currently needs to use ode integration to calculate the sliced Wasserstein loss, we currently uses 16 steps and try to reduce the computation cost related to this loss at the current stage. For future work, we are working towards more computational efficient ways.
>
> Thanks again for your time to review our paper!

---

### Official Review · Reviewer_KBvV · 2025-11-02

**Soundness:** 3
**Presentation:** 2
**Contribution:** 2
**Rating:** 2
**Confidence:** 4

**Summary:**

The paper considers unbalanced optimal transport with a velocity prior for scientific applications where such information is often available and may encode divergence-free dynamics such as rotations and curls.   The paper proposes learning both a curl-free control vector field and state-and-time dependent growth factor using a single network, in line with the Hamilton-Jacobi optimality conditions.    The authors propose several optimization tweaks to ostensibly improve training stability and convergence.

**Strengths:**

The paper proposes a natural extension of matching objectives for flows to incorporate prior guidance available in scientific applications.   While this is natural in the framing of the Schrödinger Bridge problem, the authors consider the `optimal transport with velocity prior' setting , which to my knowledge was proposed in Sec 7 of Chen, Georgiou and Pavon 2014, "On the Relation between optimal transport and Schrodinger bridges", and extend to the unbalanced case.

The paper proposes several optimization tweaks which improve training stability and convergence compared to directly solving the dual problem as in Neklyudov et. al 2023, 2024.
- HJB residual squared error loss
- temperature reweighting of HJB residual loss
- Sliced Wasserstein (intermediate) distribution matching
- endpoint total mass supervision

The authors ablate the endpoint total-mass supervision versus unbalanced AM, but several of the other choices lack ablation.

**Weaknesses:**

The clarity of the paper needs to be greatly improved.
- The velocity in the kinetic energy in Eq. 5 is not clearly specified, presumably corresponding to $v_{\text{corr}}$.
- Please move Algorithm 1 earlier in the paper.
- The simulation scheme described in Lines 143-144 should be moved until after the optimality result below Eq 8 to make clear why $\dot{x} = v_{\text{prior}}(x,t) + \nabla s(x,t)$.     The authors should also consider expanding the discussion of the fixed $p_t$ parameterization, as previous work has sought to further optimize the interpolating $p_t$ with a further neural network.
    - Algorithm 1 is unclear in specifying the samples on which we calculate the HJB residual loss.   What does sampling $[x, t, v_{\text{prior}}]$ mean?  It appears that RNA experiments may use local interval training with simulation from $x_t \sim p_{t, \text{data}}$ instead of $x_t \sim p_{t, \theta}$
    - HJB loss would benefit from explicit summation over K in the intermediate steps and and B in the endpoint expectations, as appropriate

The sliced Wasserstein distance is presented as a loss on the final distributions $\hat{\rho}_1, \rho_1$ only.   However, Algorithm 1 implements this at intermediate steps using the pushforward from the previous timestep.
- The authors should discuss the differentiability of the Sliced W2 loss.   Do we backpropagate through the ODE dynamics?  How about the weights?  I presume these reweight the W2 calculation, but what about their gradients?


The proof of Theorem 3.4 is not provided.  I might prefer that this is presented before Corollary 3.3, since subsequent choices such as squared-residuals and reweighting in Eq. 10 do not necessarily yield the desired solutions.   Thm 3.4 appears to refer to the optimality of Eq. 8.


The description of the Gaussian Translation task should be improved (what is $v_{\text{prior}}$?).   The source distribution is not clear in Figure 1.

$v_{\text{ot}}$ is used in various places (L269, L624, 644)

**Questions:**

The importance weights in the HJB residual are calculated across different time points.   One could imagine that higher residuals in particular times would lead to the $s_t$ function nat these $t$ rarely being emphasized or optimized in the residual loss.   Does this occur?   Importance weighting $w_{t,b}$ across samples at a particular time point would require aligning times within a batch, but is a different, natural choice.


Is it reasonable to expect to have supervision on the total mass in unbalanced settings?  How is this ground-truth derived/estimated from data?

---

> ### Author Response · Authors · 2025-11-20
>
> Thank you for your time and efforts to review this paper and raise important questions!
>
> 1. **More discussion on the choice of $\rho_t$**
> Thank you for highlighting this part and we will revise the script. For the scRNA-seq datasets, we follow Action Matching to use mixtures between adjacent snapshots $\rho_{t_k}^{\text{data}}$ to construct intermediate time marginals $\rho_t$. For example for the EB dataset, for $t \in [t_k, t_{k+1}]$ we sample $x_t$ from $(1-\alpha)\rho_{t_k}^{\text{data}} + \alpha\rho_{t_{k+1}}^{\text{data}}$ with $\alpha = \frac{t - t_k}{t_{k+1} - t_{k}}$. And for $v_t$, we are simply using a linear interpolation in time between each interval. Let $\tau \in [0,1]$, we have $v_{\text{prior}}(t, x) = (1 - \tau) v_k(x) + \tau v_{k+1}(x)$. Alternatively, since we already solve an ODE to compute the sliced $W_2$ loss, we could also define $\rho_t$ as the model push forward obtained by integrating $\dot x_t = v_{\text{prior}}(t,x_t) + \nabla s_\theta(t,x_t)$. Moreover as you have suggested, we could also parameterize $\rho_t$ like in [1], we think that since the single cell datasets like EB have only a few snapshots so it is coarse, it may make learning the full $\rho_t$ hard and also introduce additional complexity and more training. In comparison, the mixture-based $\rho_t$ is cheap, simple and also provides more training stability compared with using ode integration especially during the initial training stage. But we think for future works, a hybrid approach maybe an interesting direction.
>   \\
>
> 2. **On improving HJB loss**  -- Thanks for the suggestion. In the revised version, we are using this approach. For each time interval, we loop over K times to calculate the HJB loss and then take the summation to get the total HJB loss.
>
> 3. **On differentiability of sliced $W_2$** -- Algorithm 1 illustrates single cell datasets which have multiple time snapshots, so we use the multi-marginal training strategy and the sliced Wasserstein loss is calculated per time interval. For other datasets, such as the synthetic experiments with only two marginals, we simply compute sliced $W_2$ between the model terminal distribution and the target distribution at $t=1$.
>
> Regarding differentiability, in our implementation the sliced $W_2$ term is fully backpropagated through the ODE dynamics: we use odeint from torchdiffeq with gradients enabled, so the gradients of sliced $W_2$ with respect to the predicted particle positions are propagated back through the ODE solver to the model parameters. In our current experiments, we use an unweighted sliced $W_2$ loss. We only use particle positions to compute sliced $W_2$ and do not include the log-weights $\log w$ in this term. Thus, the log-weights do not receive gradients from sliced $W_2$ and are instead optimized via the mass loss.
>
>
> 4. **Clarifications on $v_{prior}$** -- For the Gaussian translation case, we define an affine prior drift as $  v_{\text{prior}}(x) =\ (\mu_1 - \mu_0) + \eta\(x - \mu_0)$ with the parameters $\mu_0 = (0,0)$, $\mu_1 = (0.5,6.0) $, $\Sigma_0 = ((2.5,0.0),(0.0, 0.3))$, $\Sigma_1 =((0.4, 0.0),(0.0, 2.2)) $ and $\eta=0.0$. We will revise the script to include proofs for Theorem 3.4  and yes this theorem refers to the optimality of Eq. 8.
>
> 5. **On importance weight s** thanks for this comment. In our revised version, we will update relevant sections to make it more clear. In our experiments for the single cell datasets, the HJB importance weights are computed and normalized
> separately for each time interval k and per batch instead of across different time snapshots. For example, for each interval $[t_k, t_{k+1}]$ we sample $t \in [t_k, t_{k+1}],$ evaluate residuals $r_{t,k,b}$, calculate the unnormalized weights $\tilde w_{t,k,b} \propto |r_{t,k,b}|^{-\tau},$ and then normalize them over the batch at that interval at a particular t: $w_{t,k,b}=\frac{\tilde w_{t,k,b}}{\frac{1}{B}\sum_{b'=1}^B \tilde w_{t,k,b'}}$, so there is no global normalization across different times and each interval contributes through an average of its own weighted residuals.
>
> 6. **On the supervision of total mass for the unbalanced case** we will also provide more details regard on this in the revised version as well. For the single cell dataset, since we neither have access to the full density $\rho_t$ nor to the absolute scale of $M(T_k)$, we use relative mass changes between time points instead by calculating the relative number of observed cells at each time point $T_k$. Specifically, we calculate the ground truth mass as $\log r_{\mathrm{target},k} = log \frac{M(T_k+1)}{M(T_k)}\approx\log\left(\frac{N_{k+1}}{N_k}\right)$,where $N_k$ is the number of cells at time point $T_k$.
>
>
> Reference
>
> [1] Neklyudov et al. "A computational framework for solving wasserstein lagrangian flows." ICML. 2024

---

### Official Review · Reviewer_C1pH · 2025-11-03

**Soundness:** 3
**Presentation:** 3
**Contribution:** 3
**Rating:** 6
**Confidence:** 4

**Summary:**

The authors propose learning velocity prior-guided Hamiltonian-Jacobi flows, where they use velocity-prior to construct a velocity-informed variant of the HJB equation. They evaluate their approach on a range of synthetic and single-cell experiments, demonstrating their method is able to learn both curl-free and divergence-free velocity field and is compatible with unbalanced optimal transport setting. Moreover, the method is parametrized by a single neural network without a need to separately train growth term or the interpolant.

**Strengths:**

* **Motivation**: The paper strongly motivates learning velocity priors to capture more complex motion patterns such as rotational and cyclical motion
* **Use of single network**: The authors model non-straight paths in a complex dynamics setting with a single neural network used to to learn both transport and growth dynamics
* **Range of experiments**: The work is evaluated on wide range of real-world and synthetic experiments

**Weaknesses:**

* **Limited Novelty**: Work claims novelty over modeling cyclical and rotational patterns by using velocity-based prior approach. Recent work on Curly Flow Matching [1] already addresses this challenge by constructing velocity-prior informed stochastic interpolant, solving Schrödinger bridge problem with non-zero drift in a simulation-free setting. While Curly Flow Matching only considers balanced-distributional transport, two works should be compared and discussed in the related work section.
* **Presentation**: There are inconsistencies in notation across section 3. After reading carefully, I understand that $v_t^*$ in line 119 corresponds to learnable $v_{corr}$, and that $\dot{x}$ and $v_{total}$ match in line 144, but this could be simplified. The same applies to $g_t(x)$ vs $g(t,x)$ or $s(t,x)$, vs $s_t(x)$ vs $s(t, x(t))$ or $v$ in equation 6, which I believe should be $v_{total}$. In the petal example when showing overlay of vector fields, I would strongly recommend separating $v_{prior}$, $v_{ot}$ and $v_{total}$ as it is very hard to distinguish them.

**Questions:**

* Could you provide computational cost comparison across baselines in table 2 to proposed method? Could you also report EMD/W2 metrics and compare to other baselines in EB experiments?
* Could you provide baseline comparison (same set of baselines as in table 2) on bone marrow single-cell data and also report EMD/W2 metrics? How do you evaluate how close to the ground truth the learnt growth dynamics (shown in figure 3) is?
* It would be very interesting to see some robustness analysis of the proposed method further tested in a toy setting. I wonder if authors tried running a simple case of velocity-prior applied in classic OT-CFM [2] or SF2M [3] matching setting with balanced-distribution transport and with sliced Wasserstein objective. Does this still allow you to learn cyclical patterns? In the balanced toy experiments where I assume Action Matching without unbalanced transport is used as an FM algorithm, have you tested this in a toy setting where $\mu_0 = \mu_1$ to show whether model can learn fully cyclical trajectories under $\omega=const$ rotational field?
* Line 258 mentions work by Sun et al (2025) [4] which also uses single network to train velocity field and growth dynamics. Could you expand this comparison and explain in which cases VRUOT would fail due to the lack of velocity priors? As mentioned above, it would be great to include empirical comparison across SW/EMD/W2 as well as computational cost on EB and bone marrow examples, if possible.
* Are the weights in lines 174 to 180 same as weights in line 161? If not it would be good to add different notation to separate them. How does importance reweighting depend on the choice of temperature $\tau$? It would be good to provide an ablation study.
* Could you provide a further commentary on trade-off between using two separate networks used to model transport and growth? What is the guarantee that $s_t$ recovers correct growth dynamics in chosen experimental setting? It would be good to evaluate the quality of learnt growth dynamics in a synthetic setting and compare to baselines such as Sun et al (2025) [4]

**References**

[1] Petrović, Katarina, et al. "Curly flow matching for learning non-gradient field dynamics." arXiv preprint arXiv:2510.26645 (2025).

[2] Tong, Alexander, et al. "Improving and generalizing flow-based generative models with minibatch optimal transport." arXiv preprint arXiv:2302.00482 (2023).

[3] Tong, Alexander, et al. "Simulation-free schr\" odinger bridges via score and flow matching." arXiv preprint arXiv:2307.03672 (2023).

[4] Sun, Yuhao, et al. "Variational Regularized Unbalanced Optimal Transport: Single Network, Least Action." arXiv preprint arXiv:2505.11823 (2025).

---

> ### Author Response · Authors · 2025-12-03
>
> Thanks very much for your detailed feedback and suggestions!
>
> **On eval with more metrics on EB** in the revised version, we added mmd, w1 and w2 and added three more baselines: DeepRUOT, Var-RUOT and MIOflow, please see Table 3,5 and 6.
>
> **On robustness test** thanks for the suggestions! we did a robustness test on EB instead. The result is in Table 4 in the revised version. Briefly, given mild or moderate mis-specification such as adding gaussian noise or scale to the RNA velocity as our $v_{prior}$ does not affect much in terms of $w_1$. We will keep your suggested example in mind for future additional experiments.
>
> **On Var-RUOT comparison and the growth term** We added evaluation comparisons with Var-RUOT on the EB dataset, see Table 3,4,5,6. We also added comparison on methods in the related work section. Briefly even though both methods use a single network and the unbalanced OT setting, Var-RUOT uses global-in-time training strategy through SDE simulation. Our method is trained locally on a per time interval basis and we use ODE integration. Moreover, we use a mixture-based sampling strategy to interpolate data between each interval whereas Var-RUOT evaluates its objective by integrating the dynamics over the entire time horizon, which requires simulating the full trajectory on a fine time grid. This is more computational expensive and time consuming. Although, based on the comparison, Var-RUOT outperforms than most other methods and performs slightly better than ours.
>
> In our limitation section, we discussed the the limitation on the growth model which currently is not biologically plausible. For future direction, additional constraints could be added for future improvement. For more discussion, see the limitation section in the last page.
>
> **On presentation** Thanks very much for pointing this out. We have revised the notations throughout the revised version for coherence and clarity.
>
> **On recent relevant works** We have added a comparison with Curly FM in the related work section. Briefly, Curly FM uses a two-stage training which needs to learn a reference drift using RNA velocity first. In comparison, our formulation incorporates it directly without additional training and use it as a "free drift".

---

### Meta-Review · Area_Chair_iBwa · 2025-12-13

**Summary:**

Reviewer concerns centered around novelty, clarity, and relations to prior work. Specifically,
* Reviewers noted that the proposed method is relatively similar to prior works in that it involve essentially a combination of prior techniques. In my view this is not disqualifying on its own, but is a valid concern.
* Most reviewers brought up concerns around clarity and presentation. Specifically, the existence of an unproven theorem, lack of experimental details, and the notation were all noted by multiple reviewers.
* Relationship to prior works: Reviewers C1pH Hm4T and r9zW all mentioned recent work by Petrovic et al. 2025, as an instance of a related work which should be discussed.

**Reviewer Concerns:**

While the rebuttal mentions updates to the manuscript, I was unfortunately unable to find these updates, which makes it difficult to evaluate the mentioned changes. The rebuttals were uploaded quite late in the review process and as such only 1/4 reviewers engaged in discussion before the discussion freeze.

After reading the discussion carefully,
* Concerns around similarity to prior works is well addressed. I am confident that this method at least adds something to the literature in terms of framing a decomposed velocity into $v_{prior}$ and $v_{corr}$.
* Concerns around clarity and presentation. Are not fully addressed without an updated draft.
* Concerns around the relationship with prior works is partially addressed. Again, it would have been nice to see the updated discussion in the manuscript.

**Reviewer Scores:**

C1pH: Initially 6: Would remain unchanged given the rebuttal, as it is likely not strong enough resolution of the limited novelty weakness to warrant an 8.

KBvV: Initially 2: Without access to the updated manuscript it is unlikely that this reviewer would have changed score

Hm4T: Initially 2: The rebuttal is again unconvincing without the updated manuscript.

r9zW: Initially 2: The reviewer commented that "the additional clarifications and experiments have addressed part of my concerns, so I am happy to increase my score at this stage". I would say given the results presented I would have updated the score to 4 in this case.

Overall, while this work has merit, but the limited novelty combined with concerns around clarity and presentation make it difficult to fully appreciate this work, especially without an updated manuscript. For these reasons I recommend rejection at this time. I recommend the authors revise their manuscript according to the reviewers' feedback to address concerns around clarity, novelty, and experimental setting at this time.

---

### Decision · Program_Chairs · 2026-01-26

Reject